# Regulation of Autophagosome–Lysosome Fusion by Human Viral Infections

**DOI:** 10.3390/pathogens13030266

**Published:** 2024-03-20

**Authors:** Po-Yuan Ke

**Affiliations:** 1Department of Biochemistry & Molecular Biology and Graduate Institute of Biomedical Sciences, College of Medicine, Chang Gung University, Taoyuan 33302, Taiwan; pyke0324@mail.cgu.edu.tw; Tel.: +886-3-211-8800 (ext. 5115); Fax: +886-3-211-8700; 2Liver Research Center, Chang Gung Memorial Hospital, Taoyuan 33305, Taiwan

**Keywords:** autophagy, virus, autophagosome, lysosome, autophagosome–lysosome fusion

## Abstract

Autophagy plays a fundamental role in maintaining cellular homeostasis by eliminating intracellular components via lysosomes. Successful degradation through autophagy relies on the fusion of autophagosomes to lysosomes, which leads to the formation of autolysosomes containing acidic proteases that degrade the sequestered materials. Viral infections can exploit autophagy in infected cells to balance virus–host cell interactions by degrading the invading virus or promoting viral growth. In recent years, cumulative studies have indicated that viral infections may interfere with the fusion of autophagosomes and lysosomes, thus benefiting viral replication and associated pathogenesis. In this review, I provide an overview of the current understanding of the molecular mechanism by which human viral infections deregulate autophagosome–lysosome fusion and summarize the physiological significance in the virus life cycle and host cell damage.

## 1. Introduction

Autophagy is a stress-response pathway in which cytosolic components are sequestered within double membrane-bound autophagosomes, which fuse with lysosomes to form autolysosomes containing lysosomal proteases to degrade the engulfed materials [1,2,3]. Upon virus infection, host cellular autophagy can be induced to eliminate the infecting virus, thus serving as the first line of antiviral response to restrict virus growth [4,5,6,7]. Some viruses, such as herpesviruses, can suppress autophagy initiation in infected cells to counteract this host defense mechanism [8,9]. In some cases of viral infection, such as in poliovirus and coronavirus infection [10,11,12,13,14], autophagy induction by infected cells promotes the generation of double-membrane vesicles to enhance viral replication. In addition, the activation of autophagy by several viruses, including flaviviruses and enteroviruses, can facilitate virus spread through the assembly and release of infectious particles through autophagic vacuoles [15,16,17,18,19,20].

Moreover, viruses, such as hepatitis C virus (HCV) and influenza A virus (IAV), can subvert autophagy to repress innate antiviral immunity and promote viral growth [21,22,23,24,25]. Recently, accumulated evidence has demonstrated that virus infection interferes with the fusion of autophagosomes and lysosomes and inhibits the acidification of lysosomes to enhance the replication of viruses and the maturation of viral particles. In this review, I outline the current knowledge on how human viruses regulate autophagosome–lysosome fusion to promote the viral life cycle. Then, the underlying molecular mechanisms are reviewed. Finally, the physiological significance of the ability of viruses to modulate the fusion of autophagosomes and lysosomes on the pathogenicity of virus-related human diseases is discussed.

## 2. General Regulation of Autophagy

Macroautophagy (hereafter referred to as “autophagy”) is a lysosomal degradation process that involves many stages, including initial nucleation, membrane elongation, closure, the maturation of autophagosomes, fusion to lysosomes, and termination [1,2,3,26,27,28,29]. Several autophagy-related gene (ATG)-encoded proteins and associated protein complexes are involved in autophagy. When mammalian cells are stressed, such as through nutrient starvation, inhibition of mammalian target of rapamycin complex 1 (mTORC1) leads to the activation and recruitment of the unc-51 like-kinase (ULK) complex (ULK1/2, FAK family-interacting protein of 200 kD [FIP200/also named RB1CC1], ATG101, ATG13) from the cytoplasm to the proximal region of the endoplasmic reticulum (ER) membrane (Figure 1) [30,31,32,33]. Subsequently, the ULK complex induces ER translocation and the activation of the class III phosphatidylinositol-3-OH kinase (PI3KC3) complex I (PI3KC3-CI contains PI3KC3 [also known as Vps34], PI3KR4 [also known as Vps15], ATG14, and Beclin 1), thus generating phosphatidylinositol-3-phosphate (PtdIns(3)P) (Figure 1) [28,34,35,36]. PtdIns(3)P then recruits downstream effectors, including WD-repeat domain PtdIns(3)P-interacting (WIPI, the mammalian ortholog of ATG18) family proteins and double-FYVE-containing protein 1 (DFCP1), for the formation of IM/phagophores from the ER (Figure 1) [27,34,35,36,37]. Among the four WIPIs, WIPI3 and WIPI4 recruit ATG2 and facilitate lipid transport and tethering between the ER and IM/phagophore [38,39,40], while WIPI2 binds to ATG16 to elongate the IM/phagophore [34]. In addition, several lipid scramblases, including ATG9, vacuole membrane protein 1 (VMP1), and transmembrane protein 41B (TMEM41B), can interact with ATG2, promote the scrambling of phospholipids in the membrane, and balance phospholipid density after lipid transfer [41,42,43,44,45,46,47]. VMP1 and ATG9 also participate in IM/phagophore emergencies and their dissociation from the ER [48,49,50,51,52,53,54,55]. In addition to the ER, autophagy can be initiated by IM/phagophore nucleation of membrane resources provided by other intracellular membranes, including the plasma membrane, mitochondria, endosome, Golgi apparatus, and mitochondria-associated ER membrane (MAM) [56,57,58,59,60,61,62,63].

Two ubiquitin-like (UBL) conjugation systems, the ATG12-ATG5-ATG16 complex formation and ATG8/microtubule-associated protein light chain 3 (LC3) family proteins-phosphatidylethanolamine (PE)-conjugation (also named ATG8/LC3 lipidation) systems, participate in the elongation and closure of the IM/phagophore into mature autophagosomes (Figure 1) [33,64,65,66,67,68]. ATG12 is covalently conjugated to ATG5 to form an ATG12-ATG5 conjugate by activating enzyme 1, ATG7, and conjugation enzyme 2, ATG10. The ATG12-ATG5 conjugate subsequently binds to ATG16, thus forming a trimeric ATG12-ATG5-ATG16 complex [64,65,69,70,71], which acts as a ubiquitin E3 ligase for ATG8/LC3 lipidation [72]. After protein translation, the precursor forms of ATG8/LC3 family proteins are cleaved by ATG4 proteases at the C-terminus to generate ATG8/LC3s-I, which are further conjugated to PE and form ATG8/LC3s-II (also known as ATG8/LC3s-PE and lipidated ATG8/LC3s) [72,73,74,75,76]. The lipidated forms of ATG8/LC3 promote the elongation of the IM/phagophore to form enclosed and mature autophagosomes by facilitating membrane fusion [69,77,78].

Mature autophagosomes fuse with lysosomes, thereby becoming autolysosomes capable of eliminating degradative cargoes via acidic proteases (Figure 1) [68,77,78,79,80,81]. Several molecules that function in the transport of vesicles and membrane fusion have been demonstrated to control the fusion of autophagosomes and lysosomes, which is mediated by several molecules, including soluble N-ethylmaleimide-sensitive-factor attachment protein receptor (SNARE) proteins [82,83,84,85,86], the Rab family of small GTPases [87,88,89,90,91], tethering factors [83,92,93,94,95,96,97,98], and cytoskeletal motor proteins [90,91,99,100,101,102,103]. After autophagy is complete, the resupply of nutrients reactivates mTOR, suppresses autophagy initiation, and promotes autophagic lysosome reformation (ALR), thus leading to autophagy termination (Figure 1) [104]. Phosphoinositides, clathrin-mediated endocytosis-associated molecules (clathrin and adaptor protein 2 [AP2]), end-directed microtubule motors, kinesin heavy chain family protein 5B (KIF5B), lysosomal efflux permease, spinster (Spin), and the ubiquitin E3 ligase Cullin 3-Kelch-like protein 20 (KLHL20) have been shown to participate in the ALR at the termination of autophagy [105,106,107,108,109,110,111,112,113,114,115].

## 3. Molecular Regulation of Autophagosome–Lysosome Fusion

The successful fusion of autophagosomes to lysosomes is critical for the acquisition of lysosomal proteases to degrade sequestered materials. Autophagosomal membrane-associated syntaxin 17 (Stx17) associates with synaptosome-associated protein 29 (SNAP29) and late endosome/lysosome-bound vesicle-associated membrane protein 7 (VAMP7) or VAMP8, thus forming functional SNARE complexes (so-called the “STX17-SNAP29-VAMP7/VAMP8” complex) to promote membrane fusion between autophagosomes and lysosomes (Figure 2) [88,89]. In addition, another SNARE complex composed of YKT6, SNAP29, and STX7 (referred to as the “YKT6-SNAP29-STX7” complex) can non-redundantly drive the fusion of autophagosomes with lysosomes (Figure 2) [86,116]. Very recently, Zheng et al. reported that YKT6 might represent a priming role for facilitating the formation of the “STX17-SNAP29-VAMP8” complex and promoting autophagosome–lysosome fusion [117]. Posttranslational modifications (PTMs), such as the phosphorylation of STX17 at serine (Ser) 2 [85], the acetylation of STX17 at lysines 219 and 223 [118], and the O-GlcNAcylation of SNAP29 at Ser2, Ser61, threonine (Thr) 130, and Ser153 [119], may differentially interfere with formation of the SNARE complex, thereby suppressing the fusion of autophagosomes to lysosomes.

Moreover, assembly of the SNARE complex during autophagosome–lysosome fusion requires tethering factors, including the homotypic fusion and protein sorting (HOPS) complex and ATG14 (Figure 2) [87,96]. The HOPS complex is recruited to the membrane fusion site by autophagosome-localized STX17, late endosomal/lysosomal Rab7, and the Rab7-interacting protein pleckstrin homology domain-containing family M member 1 (PLEKHM1), which is also capable of binding to ATG8/LC3 family proteins on autophagosomes [92,93,120]. Through interactions with STX17 and SNAP29, autophagosome-localized ATG14 stabilizes the formation of the STX17-SNAP29-VAMP8 complex and promotes lipid and content mixing for autophagosome–lysosome fusion [83]. In addition to these two tethers, ectopic P granule protein 5 (EPG5) [95], Golgi reassembly stacking protein 55 (GRASP55, also named GORASP2) [96], tectonin β-propeller repeat containing 1 (TECPR1) [97], and BIR repeat containing ubiquitin-conjugating enzyme (BRUCE) [98] can be recruited by their interaction with autophagosomal ATG8/LC3 family proteins and Rab7 on late endosomes and lysosomes and serve as membrane tethering factors for the fusion of autophagosomes to lysosomes (Figure 2).

The minus end-directed and retrograde movement of microtubules allows autophagosomes to meet lysosomes in the perinuclear region, thus facilitating their fusion. This transport of autophagosomes is mediated by a Rab7-interacting protein (RILP) and its association with the dynein–dynactin motor complex of microtubules and interaction with LC3 on autophagosomes (Figure 2) [91,100,101]. In addition, RILP also interacts with a cholesterol sensor protein, oxysterol-binding protein (OSBP)-related protein 1L (ORP1L), to promote the minus end-directed movement of autophagosomes on microtubules (Figure 2) [102]. In addition, ORP1L also cooperates with RILP to recruit the HOPS complex via PLEKHM1 to promote autophagosome–lysosome fusion. In addition, FYVE and coiled-coil (CC) domain-containing protein (FYCO1), another Rab7 effector, can form an adaptor complex with Rab7 and LC3B to promote plus end-directed transport of autophagosomes on microtubules, presumably through the motor protein kinesin (Figure 2) [90]. Histone deacetylase-6 (HDAC6)-recruited cortactin and its consequent F-actin polymerization also participate in the fusion of autophagosomes to lysosomes [121].

In addition to Rab7, several Rab family proteins diversely regulate autophagosome–lysosome fusion (Figure 2). Rab34B and its GTPase-associated protein OATL1 (also named TBC1D25) suppress the fusion of autophagosomes to lysosomes [122], while Rab21 and Rab24 promote the fusion of autophagosomes and lysosomes via their interactions with RILP and VAMP8 [123,124,125]. Furthermore, the UV-irradiation-resistance-associated gene (UVRAG), which is a component of the PI3KC3 complex shown to interact with Beclin 1 [32], promotes the fusion of autophagosomes with lysosomes by enhancing Rab7 activity and facilitating the recruitment of GRASP55 [36,126,127]. In contrast, another component of the PI3KC3 complex, the RUN domain and cysteine-rich domain-containing Beclin 1-interacting protein (Rubicon), negatively regulates autophagy by suppressing autophagosome–lysosome fusion [36,120,127].

In addition to proteins, phosphoinositides, including PtdIns(3)P and PtdIns(4)P, can drive autophagosome–lysosome fusion (Figure 2). Autophagosomal PtdIns(3)P is required to recruit the tethering factor TECRP1 to autophagosomes [97]. In addition, PtdIns(3)P on autophagosomes can recruit and activate cortactin, thereby promoting F-actin polymerization for vesicle transport via autophagosome–lysosome fusion [128]. The GABARAP-mediated translocation of phosphatidylinositol 4-kinase IIα (PI4KIIα) from the trans-Golgi network (TGN) to autophagosomes promotes the local generation of PtdIns(4)P, thus facilitating the fusion of autophagosomes to lysosomes via an unresolved mechanism [129]. Notably, the conversion of PtdIns(4)P to PtdIns(4,5)P2 by phosphatidylinositol-4-phosphate 5-kinase ɣ (PIP5Kγ) on late endosomes and lysosomes antagonizes autophagosome–lysosome fusion by promoting the release of Rab7 and PLEKHM1 from late endosomes and lysosomes [130].

## 4. Regulation of Autophagosome–Lysosome Fusion by Human Viruses

### 4.1. Hepatitis C Virus

HCV is a positive, single-stranded RNA and enveloped virus that belongs to the genus *Hepacivirus* within the *Flaviviridae* family [131,132,133,134]. Chronic HCV infection often leads to the development of end-stage liver diseases, including liver cirrhosis and hepatocellular carcinoma (HCC) [131,132]. In the late 2000s, Sir et al. first demonstrated that the replication of HCV RNA (JFH1 strain, belonging to genotype 2a) induces the accumulation of autophagosomes in the human hepatoma cell line Huh7 (Table 1) [135]. Additionally, bafilomycin-A1 (BAF-A1) treatment fails to further increase the level of LC3-II, which is a hallmark of autophagosome formation, in HCV-infected cells compared to that in mock-infected cells, suggesting that HCV infection may block autophagic flux and interfere with the fusion of autophagosomes to lysosomes [135]. HCV-induced incomplete autophagy was required to efficiently replicate HCV viral RNA [135]. Later, Sir and colleagues showed that HCV RNA colocalizes with GFP-LC3 puncta- and endogenous LC3 puncta-labeled autophagosomes in Huh7 cells harboring a subgenomic replicon, which contains an HCV genome fragment of nonstructural (NS) 3 to 5B (Table 1) [136]. Additionally, HCV NS5A and NS5B function in the assembly of the HCV RNA replication complex, which was detected in endogenous LC3 puncta in HCV replicon cells [136]. Moreover, HCV RNA, NS5A, and NS5B are found in purified autophagosomal membranes in HCV replicon cells, supporting the notion that HCV induces the formation of autophagosomes to support the formation of the membranous compartment for the replication of viral RNA [136].

On the other hand, Wang et al. reported that HCV infection induces the protein expression of Rubicon earlier than UVRAG infection (Table 1) [137]. Genetic knockdown of Rubicon promotes the formation of RFP^+^/GFP^−^-labeled autolysosomes of the mRFP-GFP-LC3 reporter in HCV-infected cells and suppresses HCV viral replication [137]. In addition, overexpression of Rubicon delays RFP^+^/GFP^−^-labeled autolysosome maturation and promotes HCV viral replication in HCV-infected cells [137]. In contrast, ectopic expression of UVRAG in HCV-infected cells facilitates the formation of RFP^+^/GFP^−^ autolysosomes and enhances the replication of HCV viral RNA [137]. Moreover, the authors showed that HCV NS4B alone can induce Rubicon expression and promote autophagosome maturation [137]. These studies point out that HCV infection may transiently inhibit the fusion of autophagosomes to lysosomes via NS4B-mediated upregulated expression of the Rubicon protein and the suppression of UVRAG protein expression [137]. Along with this study, HCV infection-induced Rubicon expression was reported to repress autophagy and activate innate immune response [138]. Another study by Weinman’s group showed that HCV infection upregulates the protein expression of ADP-ribosylation factor-like protein 8B (Arl8b), which is an Arf-like GTPase required for the trafficking of lysosomes, leading to the redistribution of lysosomes to the peripheral region and the repression of autophagic flux (Table 1) [139]. Gene silencing of Arl8b in HCV-infected cells restores autophagic flux and suppresses the release of infectious virions [139]. These results suggest that HCV can activate Arf8b expression to block the fusion between autophagosomes and lysosomes.

### 4.2. Enteroviruses

Enterovirus (EV) is an unenveloped, positive-sense, single-stranded RNA virus belonging to the *Picornaviridae* family [140,141,142,143]. Most EV infections are asymptomatic and self-limiting but still cause severe disease manifestations, such as hand-foot-and-mouth disease and neurological disorders in infants, children, and immunodeficient persons [140,141,142,143]. Coxsackievirus B3 (CVB3) belongs to the *Enterovirus* genus within the *Picornaviridae* family and often leads to gastrointestinal distress and cardiomyopathy in infected individuals [144]. CVB3 infection interferes with the fusion of autophagosomes with lysosomes in infected cells [145]. Luo’s group showed that CVB3 infection induces the accumulation of GFP-LC3-labeled autophagosomes in the heart tissues of infected GFP-LC3 transgenic mice (Table 1) [145]. Also, CVB3 infection into HEK293 human embryonic kidney cells leads to the predominant expression of RFP^+^/GFP^+^-labeled autophagosomes of the mRFP-GFP-LC3 reporter, suggesting that CVB3 infection represses autophagic flux [145]. Moreover, the authors demonstrated that CVB3 proteinase 3C induces the cleavage of SNAP29 and PLEKHM1 at glutamine (Q) 161 and Q668, respectively, in infected cells [145]. In addition, CVB3 infection blocks the interaction between STX17 and VAMP8 [145]. Gene silencing of SNAP29 and PLEKHM1 in CVB3-infected HeLa human cervical cancer cells suppresses autophagic flux and promotes virus replication [145]. These studies indicate that CVB3 infection may inhibit the assembly of the STX17-SNAP29-VAMP8 SNARE complex and the recruitment of membrane tethers by proteinase 3C-cleaved SNAP29 and PLEKHM1, thus blocking the fusion between autophagosomes and lysosomes [145].

Meanwhile, Tian and colleagues reported that CVB3 infection promotes the formation of autophagosomes in HeLa cells in a dose-dependent manner (Table 1) [146]. Additionally, BAF-A1 treatment did not further increase the LC3-II level in CVB3-infected cells, suggesting that CVB3 infection inhibits autophagic flux [146]. CVB3 infection also leads to the accumulation of RFP^+^/GFP^+^-labeled autophagosomes and a decrease in the mRNA level of STX17 in infected cells [146]. Moreover, ectopic expression of STX7 in CVB3-infected cells results in the efficient fusion of autophagosomes with lysosomes and prevents viral-induced apoptosis of infected cells [146]. These results suggest that CVB3 may interfere with autophagosome–lysosome fusion by suppressing the gene expression of STX17 [146]. Similarly, enterovirus D68 (EV-D68), which is another EV that predominantly causes acute flaccid myelitis in infected children, has also been shown to subvert the fusion of autophagosomes and lysosomes (Table 1) [147]. EV-D68 infection in HeLa cells increases the expression of LC3-II and induces the cleavage of sequestosome 1 (SQSTM1), which is a cargo receptor that targets degradative cargoes for autophagic degradation [147]. Also, EV-D68 induces the cleavage of SNAP29 at Q161 by the 3C protease but also enhances the protein expression of STX17 in infected cells [147]. Genetic knockdown of SNAP29 inhibits virus replication and virion secretion at the early stage of infection [147]. Moreover, these authors identified another SNAP protein, SNAP47, which may analogously regulate the autophagic process in EV-D68-infected cells and promote the production of infectious intracellular viral particles [147]. These studies imply that EV-D68 may block the fusion of autophagosomes to lysosomes in a fashion similar to that of CVB3 [147].

### 4.3. Influenza A Virus

IAV is an enveloped negative-sense single-strand RNA virus belonging to the *Alphainfluenzavirus* genus within the *Orthomyxoviridae* family [148,149,150]. IAV infection leads to systematic symptoms ranging from respiratory disease symptoms, such as fever, cough, and sore throat, to muscle pain, heart failure, and lethal pneumonia [148,149,150]. Gannage and colleagues first demonstrated that IAV infection in A549 human lung epithelial cells induces the accumulation of GFP-LC3-labeled autophagosomes, which colocalize with polyubiquitinated compartments and SQSTM1 (Table 1) [151]. Additionally, IAV-triggered autophagosomes do not fuse with lysosomes in infected cells [151]. In addition, the authors showed that IAV matrix protein 2 (M2) alone can induce autophagosome accumulation by blocking the fusion of autophagosomes to lysosomes [151]. Interference with autophagy in IAV-infected cells promotes cell apoptosis and the extracellular release of viral proteins [151]. These studies point out that the IAV M2 protein may repress autophagosome–lysosome fusion [151]. In line with these findings, another study by Chanda’s group showed that IAV M2 interacts with TBC1D5/OATL1 through its cytoplasmic tail in infected HEK293T cells (Table 1) [152]. The binding of IAV M2 to TBC1D5/OATL1 promotes the targeting of IAV M2 to lysosomes, and IAV M2 interferes with the interaction between Rab7 and TBC1D5/OATL1 in infected HEK293T cells [152]. Genetic silencing of TBC1D5/OATL1 in IAV-infected A549 cells promotes virus replication and promotes virus-induced lethality in infected mice [152], suggesting that TBC1D5/OATL1 restricts IAV infection [152]. These studies indicate that IAV M2 may block the fusion of autophagosomes to lysosomes by binding to TBC1D5/OATL1 and interrupting the interaction between TBC1D5/OATL1 and Rab7 [152].

### 4.4. Severe Acute Respiratory Syndrome Coronavirus 2

Severe acute respiratory syndrome coronavirus 2 (SARS-CoV-2) is an enveloped, positive-sense, single-stranded RNA virus belonging to the genus *Sarbecovirus* within the *Coronavirus* family [153,154,155]. Most SARS-CoV-2 infections cause fever, headache, and cough in infected individuals and may lead to dyspnea and hypoxemia in severe cases [153,154,155]. Zhang’s group first demonstrated that ectopic expression of SARS-CoV-2 open reading frame 3a (ORF3a) in HeLa cells induces the accumulation of endogenous LC3 puncta, and SARS-CoV-2 ORF3a decreases the nutrient starvation-induced formation of RFP^+^/GFP^−^ autolysosomes and blocks the colocalization of LC3 and LAMP1 in starved HeLa cells (Table 1) [156]. Also, SARS-CoV-2 ORF3a was shown to colocalize with the LAMP1- and Rab7-positive late endosomes and lysosomes and interact with VPS39, which is a component of the HOPS complex, in HeLa cells [156]. In addition, SARS-CoV-2 ORF3a induces VPS39 accumulation on late endosomes and lysosomes and suppresses the interaction between the HOPS complex and autophagosomal STX17 [156]. SARS-CoV-2 ORF3a inhibits the assembly of the STX17-SNAP29-VAMP7 SNARE complex and triggers lysosomal damage [156]. Moreover, loss of SNAP29 O-GlcNAcylation by genetic silencing of O-linked β-N-acetylglucosamine (O-GlcNAc) transferase (OGT) increases starvation-induced RFP^+^/GFP^−^ autolysosome formation and restores the formation of the STX17-SNAP29-VAMP7 SNARE complex in SARS-CoV-2 ORF3a-expressing cells [156]. These results imply that SARS-CoV-2 ORF3a blocks the fusion of autophagosomes with lysosomes by interfering with the HOPS complex-mediated formation of the SNARE complex [156].

Meanwhile, Zhang et al. reported that among thirteen SARS-CoV-2 viral genome-encoded proteins, ectopic expression of ORF3a induces elevations in LC3-II and SQSTM1 protein expression in HEK293T cells (Table 1) [157]. The exogenous expression of SARS-CoV-2 ORF3a in HeLa and A549 cells similarly leads to upregulated levels of LC3-II and SQSTM1 in a dose-dependent manner, and ORF3a overexpression triggers the accumulation of GFP-LC3-labeled puncta and RFP^+^/GFP^+^ autophagosomes of a mCherry-GFP-LC3 reporter [157]. In addition, the authors showed that SARS-CoV-2 ORF3a localizes to LAMP1-labeled lysosomes in HeLa cells and that the transmembrane (TM) domain and C-terminus of SARS-CoV-2 ORF3a are both required for the induction of autophagosome accumulation [157]. Another study indicated that SARS-CoV-2 ORF3a directly binds to VPS39 via tyrosine (Y) 160 at its C-terminal region in HeLa cells, thus interrupting the interaction between VPS39 and Rab7 and coincidently disrupting the association of SNAP29 with VAMP8, which is necessary for the assembly of the SNARE complex [157]. These studies imply that SARS-CoV-2 ORF3a inhibits autophagosome–lysosome fusion by interacting with VPS39 and interfering with the formation of functional fusion machinery [157].

Similarly, Qu and colleagues showed that in a similar fashion to ORF3a overexpression, SARS-CoV-2 infection leads to an increased amount of LC3-II and SQSTM1 in HeLa-hACE2 cells, which support a complete SARS-CoV-2 life cycle and VERO-E6 African green monkey kidney cells (Table 1) [158]. The authors demonstrated that SARS-CoV-2 ORF3a specifically binds to UVRAG and interferes with the interaction between UVRAG and the PI3KC3 complex [158]. Genetic depletion of ATG3 and ATG5 dramatically inhibits the replication of SARS-CoV-2 viral RNA in infected mouse embryonic fibroblast (MEF)-hACE2 cells, while SARS-CoV-2 ORF3a overexpression enhances viral RNA replication in SARS-CoV-2-infected Calu-3 human lung adenocarcinoma cells [158]. These studies suggest that SARS-CoV-2 ORF3a targets UVRAG, thus preventing its interaction with the PI3KC3 complex and thereby repressing the fusion of autophagosomes with lysosomes [158]. Notably, SARS-CoV ORF3a is not able to inhibit autophagosome–lysosome fusion [157,158]. Recently, Zhu et al. showed that adeno-associated virus (AAV) gene delivery of SARS-CoV-2 ORF3a in mouse brains leads to neurological disturbance, neurodegeneration, and cell death (Table 1) [159]. In addition, SARS-CoV-2 ORF3a expression also induces the glial response and activates inflammatory gene expression in the brains of mice [159]. Moreover, the authors reported that ectopic expression of SARS-CoV-2 ORF3a in HeLa cells blocks autophagosome–lysosome fusion and also impairs the degradation of glycosphingolipid, suggesting that the deregulation of autolysosome maturation by SARS-CoV-2 ORF3a may interfere with sphingolipid homeostasis [159].

On the other hand, SARS-CoV-2 ORF7a was also shown to be capable of repressing autophagosome–lysosome fusion (Table 1) [160]. Hou and colleagues reported that SARS-CoV-2 infection in Vero-E6 and Caco2 human colorectal adenocarcinoma cells upregulates LC3-II protein levels and induces autophagosome accumulation [160]. Treatment with 3-methyladenine (3-MA), which is an inhibitor of PI3KC3 in SARS-CoV-2-infected Vero-E6, Caco2, and HeLa-hACE2 cells, inhibits viral RNA replication and reduces the amount of extracellular infectious particles [160]. Ectopic expression of SARS-CoV-2 ORF7a in Vero-E6, Caco2, and HeLa-hACE2 cells can upregulate LC3-II expression and induce the formation of GFP-LC3-labeled puncta and RFP^+^/GFP^+^ autophagosomes of a GFP-mCherry-LC3 reporter in HeLa cells [160]. Genetic knockdown of SARS-CoV-2 ORF7a in SARS-CoV-2-infected HeLa-hACE2 and Vero-E6 cells inhibits LC3-II upregulation and decreases the protein level of SQSTM1 [160]. SARS-CoV-2 ORF7a expression and SARS-CoV-2 infection in HeLa cells activate autophagy initiation via the mTOR-ULK1 axis but suppress autophagosome–lysosome fusion by decreasing SNAP29 expression [160]. Moreover, SARS-CoV-2 ORF7a activates caspase 3 to cleave SNAP29 at aspartate (D) 30 in HeLa cells [160]. Ectopic expression of SNAP29 decreases the protein expression of LC3-II and SQSTM1 in SARS-CoV-2 ORF7a-expressing and SARS-CoV-2-infected HeLa cells and represses SARS-CoV-2 replication in infected cells [160]. These studies imply that SARS-CoV-2 ORF7a bidirectionally regulates autophagy by triggering autophagy initiation and blocking autophagosome–lysosome fusion, thereby promoting virus replication [160].

### 4.5. Human Parainfluenza Virus

Human parainfluenza virus (HPIV) is an enveloped, negative-sense, single-stranded RNA virus belonging to the *Paramyxoviridae* family [161,162,163]. HPIV infection often causes respiratory diseases, including colds, croup, and bronchiolitis, in children and adults; in some severe cases, HPIV can ultimately progress to pneumonia [161,162,163]. Ding et al. demonstrated that HPIV3 infection in LLC-MK-2 (MK-2) monkey kidney cells and HeLa cells increases LC3-II protein expression and the number of GFP-LC3 puncta in which lysosomes do not exist (Table 1) [164]. Chloroquine (CQ) and BAF-A1, which are inhibitors of autolysosome maturation, did not increase the level of LC3 in HPIV3-infected MK-2 cells [164]. HPIV3 leads to the accumulation of RFP^+^/GFP^+^-labeled autophagosomes of the mCherry-GFP-LC3 reporter in infected MK-2 cells [164]. Interference with autophagosome formation by 3-MA and gene knockout of ATG7 inhibits the production of extracellular virions in infected MK-2 and MEF cells [164]. Among the viral proteins of HPIV3, ectopic expression of phosphoprotein (P) protein alone is sufficient to induce LC3-II upregulation, the colocalization of GFP-LC3 puncta with lysosomes, and the accumulation of RFP^+^/GFP^+^ autophagosomes of the mCherry-GFP-LC3 reporter in HeLa cells [164]. Further analysis revealed that the HPIV3 P protein directly binds to SNAP29 via the N-terminal region [164]. This interaction prevents the association of SNAP29 with STX17 rather than with VAMP8 [164]. Deletion of the N-terminal region required for SNAP29 binding in the HPIV3 P protein inhibits HPIV3 P-induced autophagosome accumulation [164]. These studies suggest that the HPIV3 P protein inhibits autophagosome–lysosome fusion by interacting with SNAP29, thus promoting the accumulation of autophagosomes to produce infectious viruses [164].

### 4.6. Other Viruses

In addition to the viruses mentioned above, other viruses, such as Epstein–Barr virus (EBV) and Kaposi’s sarcoma-associated herpesvirus (KSHV), have been shown to modulate the fusion of autophagosomes with lysosomes [165,166]. In EBV-producing cells, induced autophagosomes do not colocalize with lysosomes, which coincides with the downregulation of Rab7 protein expression, thus preventing the degradation of EBV and enhancing virus replication (Table 1) [165]. Similarly, the activation of the KSHV lytic cycle induces the formation of autophagosomes, which cannot colocalize with lysosomes [166]. A reduction in Rab7 also mediates this blockade of autophagosome–lysosome fusion via KSHV lytic cycle activation [166]. These studies indicate that oncogenic viruses may deregulate the fusion of autophagosomes to lysosomes, thus promoting their escape from autophagic degradation and switching the latent–lytic infection cycle.

**Table 1 pathogens-13-00266-t001:** Regulation of autophagosome–lysosome fusion by human viruses.

Virus	Effects on the Fusion of Autophagosomes with Lysosomes	References
HCV	1. Replication of viral RNA activates incomplete autophagy. 2. HCV viral RNA replication inhibits autophagic flux and induces autophagosome accumulation. 3. The viral-mediated inhibition of autophagy promotes virus replication.	[135]
HCV	1. In HCV subgenomic replicon cells, viral RNA, NS5A, and NS5B localize on autophagosomal membranes. 2. The HCV-induced autophagosomes support the membranous compartments that are required for replicating HCV RNA.	[136]
HCV	1. HCV infection increases the protein level of Rubicon earlier than that of UVRAG. 2. HCV NS4B alone can upregulate Rubicon expression and inhibit the protein expression of UVRAG. 3. HCV-induced Rubicon expression activates innate immunity and inhibits HCV replication.	[137,138]
HCV	1. HCV infection increases Arl8b protein expression. 2. HCV infection leads to the redistribution of lysosomes. 3. HCV infection suppresses autophagic flux by increasing the Arl8b level.	[139]
CVB3	1. CVB3 infection induces autophagosome accumulation in vivo and in vitro. 2. CVB3 proteinase 3C cleaves SNAP29 and PLEKHM1. 3. CVB3 infection disrupts the interaction between STX17 and VAMP8, thus repressing autophagosome–lysosome fusion.	[145]
CVB3	1. CVB3 infection leads to the accumulation of autophagosomes. 2. CVB3 infection inhibits STX17 transcription, thus downregulating STX17 protein expression. 3. CVB3 may repress STX17 expression to block autophagosome–lysosome fusion.	[146]
EV-D68	1. EV-D68 infection upregulates LC3-II expression and induces the cleavage of SQSTM1. 2. EV-D68 induces the cleavage of SNAP29 at Q161. 3. EV-D68 infection interferes with autophagosome–lysosome fusion by cleaving SNAP29.	[147]
IAV	1. IAV infection induces autophagosome accumulation. 2. IAV M2 protein alone is sufficient to disrupt the fusion of autophagosomes to lysosomes.	[151]
IAV	1. IAV M2 interacts with TBC1D5/OATL1, leading to the lysosomal translocation of IAV M2. 2. The binding of TBC1D5/OATL1 to IAV M2 blocks its interaction with Rab7. 3. IAV M2 interferes with autophagosome–lysosome fusion through binding to TBC1D5/OATL1.	[152]
SARS-CoV-2	1. SARS-CoV-2 ORF3a interferes with autolysosome maturation and induces autophagosome accumulation. 2. SARS-CoV-2 ORF3a blocks the interaction between autophagosomal STX17 and the HOPS complex. 3. SARS-CoV-2 ORF3a interferes with autophagosome–lysosome fusion by interrupting the HOPS complex-mediated assembly of the STX17-SNAP29-VAMP7 SNARE complex.	[156]
SARS-CoV-2	1. SARS-CoV-2 ORF3a upregulates the expression of LC3-II and SQSTM1 and induces autophagosome accumulation. 2. SARS-CoV-2 ORF3a directly binds to VPS39 and suppresses the interaction between VPS39 and Rab7. 3. SARS-CoV-2 ORF3a interferes with the interaction between SNAP29 and VAMP8, thus blocking autophagosome–lysosome fusion.	[157]
SARS-CoV-2	1. SARS-CoV-2 infection induces increases in LC3-II and SQSTM1 levels. 2. SARS-CoV-2 ORF3a binds to UVRAG and blocks the association of UVRAG with the PI3KC3 complex. 3. SARS-CoV-2 infection may disrupt autophagosome–lysosome fusion by ORF3-targeting UVRAG.	[158]
SARS-CoV-2	1. SARS-CoV-2 ORF3a interferes with autophagosome–lysosome fusion and induces autophagosome accumulation in the brains of mice. 2. SARS-CoV-2 ORF3a causes neurological disturbance, neurodegeneration, and cell death. 3. SARS-CoV-2 ORF3a deregulates sphingolipid homeostasis.	[159]
SARS-CoV-2	1. SARS-CoV-2 infection and overexpression of SARS-CoV-2 ORF7a induces an increase in LC3-II levels and autophagosome accumulation. 2. SARS-CoV-2 ORF7a induces caspase 3 activation to cleave SNAP29 at D30. 3. SARS-CoV-2 ORF7a blocks autophagosome–lysosome fusion by promoting the caspase 3-mediated cleavage of SNAP29.	[160]
HPIV3	1. HPIV3 infection and overexpression of HPIV3 P protein increase the level of LC3-II and induce autophagosome accumulation. 2. HPIV3 P protein binds to SNAP29, thereby preventing the interaction between SNAP29 and STX17. 3. HPIV3 suppresses autophagosome–lysosome fusion by the P protein, which disrupts the binding of STX17 to SNAP29.	[164]
EBV	EBV blocks autophagosome–lysosome fusion by downregulating Rab7 expression.	[165]
KSHV	KSHV interferes with the fusion of autophagosomes to lysosomes by reducing the protein level of Rab7.	[166]

## 5. Conclusions and Perspectives

In recent years, several viruses have been shown to inhibit the fusion of autophagosomes with lysosomes, thus resulting in the accumulation of autophagosomes in which viruses grow. Most of these viruses interrupt the assembly of the SNARE complex via protein-mediated competitive binding, the downregulation of expression, and the proteolysis of molecules involved in autophagosome–lysosome fusion. So far, the detailed mechanism underlying this regulation remains debated and needs further investigation. For example, the spatiotemporal control of the timely actions of viral proteins in the fusion process is unclear. Notably, whether these viruses interfere with the fusion of autophagosomes to lysosomes is still debated, at least for HCV, as several studies have shown that viral-induced autophagy progresses to autolysosome maturation and is used for the turnover of organelles [21,167,168]. Little is known about how infected cells deal with the large population of autophagosomes when viruses repress their fusion to lysosomes, and whether this deregulation of fusion contributes to viral-induced pathogenesis via the dysfunction of autophagic degradation is also unclear. Most importantly, whether and how viruses interfere with autophagosome–lysosome fusion in physiologically relevant cell contexts, such as in small animal models of infection and in the tissues of individuals infected with viruses, remains largely unknown, as most past studies have used cell culture infection as a research model. In perspective, further investigations are needed to elucidate the molecular mechanism by which viruses block the fusion of autophagosomes with lysosomes and to explore the physiological significance of the dysregulation of autophagosome–lysosome fusion in virus-associated diseases in humans.

## Figures and Tables

**Figure 1 pathogens-13-00266-f001:**
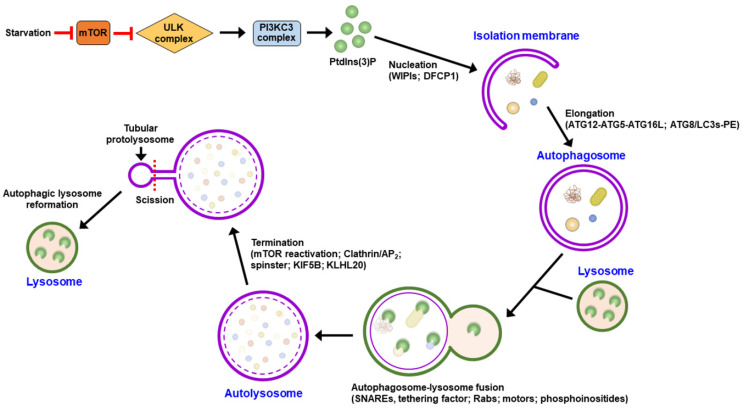
Schematic diagram of the autophagy process. Stresses, such as starvation, inhibit the activity of mTOR, thus promoting ER translocation of the ULK complex to recruit the PI3KC3 complex. The activated PI3KC3 complex generates PtdIns(3)P to recruit DFCP1 and WIPIs for isolation membrane (IM) nucleation. Elongation of the IM/phagophore to a closed autophagosome involves the ubiquitin-like conjugation system for the formation of ATG12-ATG12-ATG16L and ATG8/LC3s-PE conjugates. The autophagosome subsequently fuses with a lysosome to form an autolysosome, in which the cytosolic components are degraded. The SNARE complex, tethers, Rab family proteins, and cytoskeletal motors facilitate autophagosome–lysosome fusion. When autophagy is terminated, the tubular proto-lysosomes emerge from autolysosomes, promoting the regeneration of lysosomes.

**Figure 2 pathogens-13-00266-f002:**
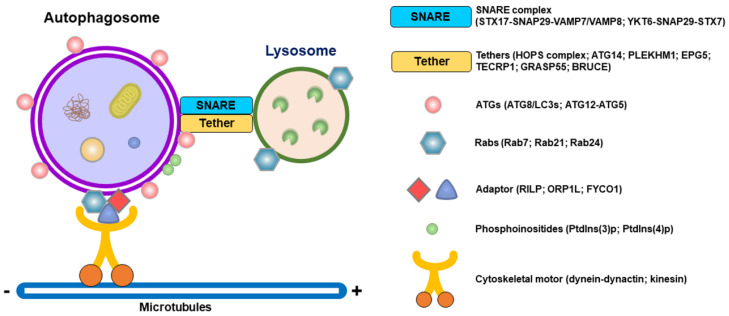
Specific molecules function in the fusion of autophagosomes with lysosomes. Various functional molecules, including the SNARE complex, tethering factors, Rab family proteins, cytoskeletal motors, autophagosomal proteins, phosphoinositides, and adaptor proteins, are required for autophagosome–lysosome fusion.

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
