# Peer review of "Regulation of Autophagosome–Lysosome Fusion by Human Viral Infections"

_pathogens, 2024, doi:10.3390/pathogens13030266_

Round 1

Reviewer 1 Report

Comments and Suggestions for Authors

In this review the author described the viral infections can exploit autophagy in infected cells to balance virus-host cell interactions by degrading the invading virus or promoting viral growth. In recent years, cumulative studies have indicated that viral infections may interfere with the fusion of autophagosomes and lysosomes, thus benefiting viral replication and associated pathogenesis. In this review, I provide an overview of the current understanding of the molecular mechanism by which human viral infections deregulate autophagosome-lysosome fusion and summarize the physiological significance in the virus life cycle and host cell damage.

The information are looking good, however there are critical concerns that need to be addressed by the authors before publication.

Major comments

1: Include evidences of HIV with autophagy (how they affect autophagy), in literature as well in the table.

2: Add one more figure about the representation of Autophagy with all these viruses mentioned included HIV.  

Author Response

Dear reviewer:

Thank you for giving me the opportunity to resubmit my manuscript “Regulation of Autophagosome-Lysosome Fusion by Human Viral Infections” to Pathogens (Manuscript ID: pathogens-2902677). I appreciate the thoughtful and constructive suggestions provided by the reviewers. The content of this manuscript has been improved based on the reviewers’ comments. The changes are shown in the revised manuscript, and point-by-point responses to each comment are listed below.

Point 1: Include evidences of HIV with autophagy (how they affect autophagy), in literature as well in the table.

Response 1: I thank the reviewer for the thoughtful comments on our manuscript. I have incorporated the studies showing how HIV infection alters autophagosome-lysosomes in section 4.6 in the revised manuscript. Please see line 397 of paragraph 3 on page 12 to line 438 of paragraph 1 on page 13, Table 1 on pages 7~8, and Figure 3 on page 9 in the revised manuscript. 

Point 2: Add one more figure about the representation of Autophagy with all these viruses mentioned included HIV.

Response 2: Thank you for your thoughtful suggestions. In the revised manuscript, I have added Figure 3 to show how human viruses interfere with autophagosome-lysosome fusion. Please see Figure 3 on page 9 in the revised manuscript.

Reviewer 2 Report

Comments and Suggestions for Authors

Review by Ke (pathogens-2902677) entitled “Regulation of autophagosome-lysosome fusion by human viral infections” described advances in the research field on host-virus interaction regarding viral subversion of autophagy processes to benefit viral propagation. Preceded by comprehensive description of macroautophagy process the author summarized evidence in the selected human viruses that autophagosome-lysosome fusion was regulated during the infection process. The manuscript pointed out important aspects of viral regulation of host cell machinery in depth, though this reviewer found it difficult to read and had to spent some time to comprehend what the author intended to describe. Perhaps it is because the text was not clear enough to elaborate ideas what author intended to address. For example, the text started with explaining the viral interference of autophagosome-lysosome fusion process without clarifying what is this to do with the viral propagation. The consequences appeared to be important for securing autophagosome that served as viral replication factory, which could have been mentioned in the Abstract. Even though it was mentioned in Conclusions and perspectives, many readers could be at a loss without such guide. This reviewer feels that the manuscript requires minor modifications to be more acceptable to readers.

Point

In Table 1, it is not clear what would be explained by compiling experimental evidence. Many listings did not correlate with the “Effects on the fusion of autophagosomes with lysosomes”. They were more like “Evidence showing association of viral replication with autophagy process”. It would be more appealing if it was organized with succinct description.

Author Response

Dear reviewer:

Thank you for giving me the opportunity to resubmit my manuscript “Regulation of Autophagosome-Lysosome Fusion by Human Viral Infections” to Pathogens (Manuscript ID: pathogens-2902677). I appreciate the thoughtful and constructive suggestions provided by the reviewers. The content of this manuscript has been improved based on the reviewers’ comments. The changes are shown in the revised manuscript, and point-by-point responses to each comment are listed below.

Point 1: In Table 1, it is not clear what would be explained by compiling experimental evidence. Many listings did not correlate with the “Effects on the fusion of autophagosomes with lysosomes”. They were more like “Evidence showing association of viral replication with autophagy process”. It would be more appealing if it was organized with succinct description.

Response 1: I am very grateful for the reviewer’s thoughtful comment. In the revised manuscript, I have reorganized and revised the content of Table 1 to summarize the effects of human viruses on autophagosome-lysosome fusion and the physiological significance of these effects on the growth of infecting viruses. Please see Table 1 on pages 5~8 in the revised manuscript.

Reviewer 3 Report

Comments and Suggestions for Authors

Po-Yuan Ke presents a thorough review on the “Regulation of autophagosome-lysosome fusion by human viral infection”. The author summarizes in the first part the autophagic process with a particular emphasis on the fusion of autophagosomes and lysosomes. In the second part, the author highlights the current knowledge on viruses that affect autophagosome-lysosome fusion. 

The organization of the review is logic, it integrates the necessary detail and is written very sound. This reviewer enjoyed reading this manuscript very much! 

One minor point that should be improved: Although the literature is not consistent, the author should be precise in terming LC3. There are three members in this subclass of the ATG8 protein family. Usually, LC3B has been detected or LC3B-consructs had been used.

Comments on the Quality of English Language

The text is very sound and the English language is fine.

Author Response

Dear reviewer:

Thank you for giving me the opportunity to resubmit my manuscript “Regulation of Autophagosome-Lysosome Fusion by Human Viral Infections” to Pathogens (Manuscript ID: pathogens-2902677). I appreciate the thoughtful and constructive suggestions provided by the reviewers. The content of this manuscript has been improved based on the reviewers’ comments. The changes are shown in the revised manuscript, and point-by-point responses to each comment are listed below.

Point 1: One minor point that should be improved: Although the literature is not consistent, the author should be precise in terming LC3. There are three members in this subclass of the ATG8 protein family. Usually, LC3B has been detected or LC3B-consructs had been used.

Response 1: I thank the reviewer for the thoughtful comments. In the revised manuscript, I have revised the terming of LC3 in several studies, which indicated that the specific form of LC3 was assessed, mostly for LC3B. However, a few studies did not specifically show what kind of LC3 was analyzed. I retained the term “LC3” in the revised manuscript. For the GFP-LC3 and mRFP-GFP-LC3 reporter constructs, these two reporters were initially constructed with rat LC3 (by Tomatsu Yoshimori’s group; please refer to PMID: 11060023 [EMBO J. 2000 Nov 1. 19(21):5720-8]; PMID: 17534139 [Autophagy. 2007;3(5):452-60]). The mCherry-GFP-LC3B reporter was constructed by Jayanta Debnath’s group (please refer to PMID: 19148225 [EMBO Rep. 2009 Feb;10(2):173-9]). Please see lines 252 and 262 on page 9; lines 302 and 303 on page 10; lines 323, 334, 356, 363, and 368 on page 11; lines 379, 381, 383, 386, 388, 405, 408, 412, 413, and 414 on page 12; and lines 428 and 435 on page 13.

Round 2

Reviewer 1 Report

Comments and Suggestions for Authors

The author has responded all the comments, and I am pleased to inform you that your review article Title: Regulation of Autophagosome-Lysosome Fusion by Human Viral Infections (Manuscript ID: pathogens-2902677) is potentially acceptable. I appreciate your efforts.

Best of luck